# Toxicity and Sublethal Effects of Autumn Crocus (*Colchicum autumnale*) Bulb Powder on Red Imported Fire Ants (*Solenopsis invicta*)

**DOI:** 10.3390/toxins12110731

**Published:** 2020-11-21

**Authors:** Sukun Lin, Deqiang Qin, Yue Zhang, Qun Zheng, Liupeng Yang, Dongmei Cheng, Suqing Huang, Jianjun Chen, Zhixiang Zhang

**Affiliations:** 1Key Laboratory of Natural Pesticide and Chemical Biology of the Ministry of Education, South China Agricultural University, Guangzhou 510642, China; zhangli0810@stu.scau.edu.cn (S.L.); peakqin@163.com (D.Q.); zy3333@stu.scau.edu.cn (Y.Z.); zhqun423@stu.scau.edu.cn (Q.Z.); yang15136283877@126.com (L.Y.); 2Department of plant protection, Zhongkai University of Agricultural and Engineering, Guangzhou 510225, China; zkcdm@zhku.edu.cn; 3College of Chemistry and Chemical Engineering, Zhongkai University of Agriculture and Engineering, Guangzhou 510225, China; hsuqing@zhku.edu.cn; 4Department of Environmental Horticulture and Mid-Florida Research and Education Center, Institute of Food and Agricultural Sciences, University of Florida, Apopka, FL 32703, USA

**Keywords:** autumn crocus, *Colchicum autumnale*, colchicine, invasive species, pest control, red imported fire ant (RIFA)

## Abstract

Autumn crocus (*Colchicum autumnale* L.) is a medicinal plant as it contains high concentrations of colchicine. In this study, we reported that the ground powder of autumn crocus bulb is highly toxic to invasive *Solenopsis invicta* Buren, commonly referred to as red imported fire ants (RIFAs). Ants fed with sugar water containing 5000 mg/L of bulb powder showed 54.67% mortality in three days compared to 45.33% mortality when fed with sugar water containing 50 mg/L of colchicine. Additionally, the effects of short-term feeding with sugar water containing 1 mg/L of colchicine and 100 mg/L of autumn crocus bulb powder were evaluated for RIFAs’ colony weight, food consumption, and aggressiveness, i.e., aggregation, grasping ability, and walking speed. After 15 days of feeding, the cumulative colony weight loss reached 44.63% and 58.73% due to the sublethal concentrations of colchicine and autumn crocus bulb powder, respectively. The consumption of sugar water and mealworm (*Tenebrio molitor* L.) was substantially reduced. The aggregation rates decreased 48.67% and 34.67%, grasping rates were reduced to 38.67% and 16.67%, and walking speed decreased 1.13 cm/s and 0.67 cm/s as a result of the feeding of the two sublethal concentrations of colchicine and autumn crocus bulb powder, respectively. Our results for the first time show that powder derived from autumn crocus bulbs could potentially be a botanical pesticide for controlling RIFAs, and application of such a product could be ecologically benign due to its rapid biodegradation in the environment.

## 1. Introduction

*Solenopsis invicta* Buren, commonly known as the red imported fire ant (RIFA), is a dominant and detrimental invasive species [1,2,3]. RIFA is native to South America [4], and was discovered in southern Alabama in the 1930s [5] and subsequently in Australia [6,7], New Zealand [8], Malaysia [9], Taiwan [10,11], and mainland China [12]. A mature RIFA colony contains a single queen or multiple queens with a large number of workers ranging from 200,000 to 400,000 [13]. They are extremely aggressive and voracious feeders and can quickly disrupt ground fauna, causing the imbalance of specific natural ecosystems [14]. Fire ant workers sting relentlessly when their mound is disturbed. To control RIFAs, many contact insecticides, including chlorpyrifos and pyrethroid, have been widely used [15]. However, the use of synthetic insecticides may cause environmental contamination and are also prone to insecticide resistance. Thus, naturally occurring, low-resistant, and ecologically friendly control methods have been explored for RIFA management [16,17]. Among them, biological pesticides or biopesticides are valuable alternatives to synthetic pesticides [18].

Biopesticides are natural products derived from living organisms including microbes, nematodes, and plants that can limit or reduce pest populations [19,20]. Biopesticides are usually unique and complex and have different modes of action; thus, pests are less likely to develop resistance. Biopesticides are generally safer than synthetic ones to the environment as they can be quickly degraded, and they are also more friendly to nontarget organisms [21,22]. The use of biopesticides has been increasing globally each year. North America is the largest biopesticide market, accounting for 44% of the market value [23].

Autumn crocus (*Colchicum autumnale* L.), also known as meadow saffron, is an herbaceous perennial plant in the family Colchicaceae and is cultivated as an ornamental plant [24,25] and medicinal herb [26]. Its flowers arise from underground bulbs without leaves and often achieve a height of 20 to 25 cm (Figure 1A). All organs of the plant are toxic and contain alkaloids, of which colchicine contributes 50–70% of the total alkaloid content [25]. These alkaloids withstand storage, drying, and heat treatment. Colchicine is a medication used to treat gout and Bechet’s disease [27,28]. It can inhibit microtubule formation in cell mitosis, thus inducing polyploid plants [29,30]. However, autumn crocus and colchicine have been rarely reported to have any insecticidal activities.

In our studies with RIFAs, we found that autumn crocus had toxic effects on RIFAs. Thus, this study was intended to analyze colchicine levels in autumn crocus bulb and determine toxic and sublethal effects of colchicine and autumn crocus bulb powder on RIFAs, including the mortality, colony weight, food consumption, i.e., sugar water and mealworm (*Tenebrio molitor*), and aggressiveness (aggregation, grasping ability, and walking speed). Our study for the first time demonstrated that autumn crocus bulbs could be considered as a biopesticide for effectively controlling RIFAs.

## 2. Results

### 2.1. Colchicine Content in Autumn Crocus Bulb 

Standard colchicine solutions (0.1 to 10 mg/L) were analyzed by HPLC, which resulted in an equation of *y* = 186,423*x* + 8971.9 (*R*^2^ = 0.9989) for determination of colchicine concentrations in autumn crocus bulb powder. The HPLC prolife showed that colchicine standard presented a single peak at the retention time of 10 min (Figure 1E), while four peaks appeared in analysis of the crocus bulb powder, and the first two were joined, the third and the fourth were independent (Figure 1F). The fourth peak had the same retention time as the colchicine standard solution (Figure 1E,F). To further confirm if the fourth peak was colchicine, HPLC-UV-Vis analysis was performed. The result showed that the fourth peak had the same UV-Vis spectra as the peak of colchicine, suggesting that it was colchicine. The third peak also had the similar UV-Vis spectra to colchicine (data not shown), it could be colchicoside. The first two peaks were not determined. Based on the equation mentioned above, colchicine content in autumn crocus bulb powder was found to be 0.30% (*n* = 3). 

### 2.2. Toxicity of Colchicine and Autumn Crocus Bulb Powder to S. invicta 

The death of RIFAs occurred after three days of feeding with sugar water containing different concentrations of colchicine (1 to 50 mg/L) or autumn crocus bulb powder (100 to 5000 mg/L), and mortality rates varied significantly with respective concentrations (Figure 2). A mortality rate of 45.33% was evident when RIFAs were fed with a solution containing colchicine at 50 mg/L; the mortality rates gradually declined with the concentration decrease and became only 5.33% at the concentration of 1 mg/L (Figure 2A). When RIFAs were fed with a solution containing autumn crocus bulb powder at 5000 mg/L, the mortality rate was 54.67%. With the reduction of bulb powder content in the solutions, the mortality rates decreased. The mortality was 6.67% at the bulb powder concentration of 100 mg/L (Figure 2B). On the contrary, the mortality rate of the RIFAs fed with sugar water solution only was 3.33%.

A sublethal concentration of colchicine (1 mg/L) and autumn crocus bulb powder (100 mg/L) in 10% sugar water was selected for testing sublethal effects on RIFAs over 15 days of observation. There were no significant differences in mortality rate among treatments on day 3, but significant difference occurred between the control and colchicine and crocus bulb powder solution treatments on day 6 (Figure 3). Subsequently, mortality rates of RIFAs fed with colchicine and crocus powder solutions became significantly higher than control after 6 days of feeding. Furthermore, the mortality rates of RIFAs fed with crocus bulb powder solution became significantly greater than those fed with colchicine from days 9 to 15. On day 15, the mortality of ants receiving 100 mg/L autumn crocus bulb powder solution increased to 52.67%, which was significantly higher than those receiving 1 mg/L colchicine in sugar water (33.33%) and the control group (9.33%) (Figure 3).

### 2.3. Effect of Colchicine and Autumn Crocus on Colony Growth of S. invicta

Solutions made from either colchicine or autumn crocus bulb powder at sublethal concentrations had significantly negative impacts on the survival of *S. invicta* colonies. The weight loss of colonies receiving 10% sugar water containing 100 mg/L autumn crocus bulb powder or 1 mg/L colchicine was significantly greater than that of the control (10% sugar water only) on day 3 (Figure 4). From days 6 to 15, the weight loss of colonies fed with 1 mg/L colchicine and 100 mg/L crocus bulb powder solutions quickly increased. Colonies fed with 100 mg/L crocus bulb powder lost significantly greater weight than those fed with 1 mg/L colchicine, whereas the colchicine treatment caused significantly higher weight loss than the control. After 15 days of feeding, the weight loss percentage of colonies fed with 100 mg/L autumn crocus bulb powder and 1 mg/L colchicine solutions were 58.73% and 44.63%, respectively, compared to 9.57% of the control. 

### 2.4. Effects of Colchicine and Autumn Crocus Bulb Powder on Food Consumption of S. invicta

The consumption of sugar water and *T. molitor* (mealworm) by colonies of RIFAs was examined. At the beginning of the experiment, per gram of ants in respective colonies consumed 178.67 mg sugar water, 164.33 mg sugar water containing 100 mg/L autumn crocus bulb powder, and 171.67 mg sugar water containing 1 mg/L colchicine (Figure 5A). There were no significant differences in the amount of sugar water consumption among the three treatments on day 1. On day 3, the amount of sugar water consumed by ants in the colonies fed with colchicine and autumn crocus bulb was similar but significantly less than that consumed by ants in the control colonies. Starting from day 6, the amount of sugar water consumed by ants fed with three food sources became significantly different, and the differences continued to day 15. The sugar water consumption by ants in the control colonies was 103.67 mg on day 15, which was significantly higher than that consumed by ants in the other two treatments colonies. The sugar water consumption by the ants receiving 100 mg/L autumn crocus bulbs was 14.67 mg on day 15, which was significantly lower than that consumed by ants fed with 1 mg/L colchicine (45.67) (Figure 5A).

A different pattern was observed in food consumption when colonies were fed *T. molitor* (Figure 5B). Ants in the colonies treated with 100 mg/L autumn crocus bulb powder and 1 mg/L colchicine solutions consumed 47.67 mg and 55.33 mg of *T. molitor*, respectively, and ants in the control colonies consumed 57.33 mg of *T. molitor* on the first day. There were no significant differences in consumption of *T. molitor* among the three treatments on day 1. On the third day, the consumption of *T. molitor* increased greatly. Ants in the colonies treated with 100 mg/L autumn crocus bulb powder and 1 mg/L colchicine solutions consumed 99.33 mg and 108.67 mg of *T. molitor*, respectively, and ants in the control treatment consumed 137.33 mg of *T. molito*. The amount of *T. molitor* consumed by ants in the colonies treated with colchicine and autumn crocus bulb did not differ significantly, but ants in the control group consumed significantly more *T. molitor* than the two other treatments. Ants in the control colonies continuously consumed more *T. molitor* until day 6, and then their consumption gradually decreased until day 15. The consumption of *T. molitor* by ants in colonies treated with colchicine and autumn crocus bulb powder decreased starting from day 3 until day 15. On day 15, per gram of ants in the colonies treated with 100 mg/L autumn crocus bulb powder and 1 mg/L colchicine consumed 39.33 mg and 65.67 mg of *T. molitor*, respectively, which was significantly lower than 104.33 mg consumed by ants in the control colonies (Figure 5B).

### 2.5. Effects of Colchicine and Autumn Crocus on the Aggressiveness of S. invicta

The aggressiveness of RIFAs was evaluated in terms of their aggregation, grasping ability, and walking speed (Figure 6). There were no differences in aggregation among workers fed with sugar water and sugar water containing 1 mg/L colchicine and 100 mg/L autumn crocus bulb powder on day 1 of the experiment (Figure 6A). From day 3, the aggregation rate of the control treatment was significantly higher than the other two treatments. Starting from the day 6 to day 15, the aggregation rates of small workers treated with autumn crocus bulb powder became the lowest, followed by the treatment of colchicine and then the control. On day 15, the aggregation rate of workers receiving sugar water containing 100 mg/L autumn crocus bulb powder was only 34.67% compared to 48.67% for the colchicine treatment and 88.67% for the control. 

The grasping rates of workers fed with the three food sources exhibited the same pattern as the aggregation rates during the 15 days of evaluation (Figure 6B). No differences were recorded on day 1, but differences occurred on day 3 where the control treatment showed significantly greater grasping rate than the rest of treatments. Significant differences among the three treatment groups occurred from day 6 to 15. On day 15, the grasping rate of workers fed 100 mg/L autumn crocus bulb powder dropped to 16.67%, which was significantly lower than 38.67% of the 1 mg/L colchicine treatment and 89.33% of the control treatment groups.

The walking speed of small workers is presented in Figure 6C. There were no significant differences in walking speed among small ants fed with the three food sources at the beginning of the experiment. On day 3, the walking speed of small workers fed with autumn crocus bulb powder solution was significantly slower than those fed with the two other sources. However, starting on day 6, the walking speed of small workers fed with sugar water only was significantly faster than those fed with colchicine and autumn crocus bulb powder solutions. From day 9 to day 15, the walking speed of small workers receiving 100 mg/L autumn crocus bulb powder solution was the lowest, followed by those receiving the treatment of 1 mg/L colchicine treatment, and then the control. On day 15, the walking speed of small workers fed with 100 mg/L autumn crocus bulb powder was reduced to 0.67 cm/s, which was significantly lower than that of small workers receiving 1 mg/L colchicine sugar water (1.13 cm/s) and the control treatment (1.97 cm/s).

## 3. Discussion

As an aggressive invasive species, *S. invicta* causes major problems, including displacement or elimination of native ground fauna and negative effects on human health and public safety [31,32]. Lard et al. [33] estimated that the cost for control of RIFAs in the United States alone was greater than USD 6 billion annually. *S. invicta* specimen was first identified in Wuchuan, Guangdong Province, China on September 23, 2004 [34]. It was reported that *S. invicta* forage activities occur year-round, peaking in the summer and fall in South China. The expansion rate of *S. invicta* in China ranged from 26.5 to 48.1 km/yr [35]. Currently, *S. invicta* has been found in more than 10 provinces in China [35]. Lin et al. [36] predicted *S. invicta* could potentially threaten 41 species on the China National List of Protected Wildlife. Due to its invasion, the abundance of native ant species decreased more than 30% in China [37].

Common methods for control of RIFAs include the use of insecticides, such as chlorpyrifos and pyrethroid or biological control agents, such as the decapitating phorid fly, *Pseudacteon tricuspis* and a pathogen of fire ants, *Thelohania solenopsae* [38]. Additionally, plant-based chemicals have also been explored. For example, essential oil extracted from leaves of cinnamon (*Cinnamomum osmophloeum*) and trans-cinnamaldehyde were found to have an inhibitory effect on RIFAs [39]. As far as is known, there has been no report on the use of autumn crocus bulb powder for RIFA control. Our results showed that when RIFAs were fed a sugar water solution containing autumn crocus bulb powder at 5000 mg/L, the mortality rate reached 54.67% in three days, which was higher than 45.33% when fed 50 mg/L colchicine (Figure 2). Since the autumn crocus bulbs are easy to produce, and powder can be quickly ground, the use of autumn crocus bulb powder could be a simple and convenient way for effective control of RIFAs.

Autumn crocus has been used as an approved medical plant for over 3000 years even though it is poisonous [40]. Approximately 30 tropolone alkaloids have been identified in *C. autumnale*, of which colchicine, demecolcine, and colchicoside are the most abundant [41]. The concentration of colchicine ranged from 0.12 to 1.9% in bulb, 0.02 to 1.42% in leaves, 0.15 to 0.85% in flowers, and 0.14 to 1.2% in seeds. Demecolcine varied from 0.18% to 0.37% in bulb, and about 0.08% in leaves; however, colchicoside occurs mainly in seeds ranging from 0.48 to 0.1% [41]. In the present study, colchicine content in crocus bulb powder was 0.3%. Using solutions made from the powder, along with colchicine solution as a positive control, RIFAs exhibited almost identical patterns in response to both colchicine and autumn crocus bulb powder solutions. Our results suggest that the mortality of RIFAs is caused by the action of colchicine in autumn crocus bulb powder. 

The sublethal concentrations of colchicine and autumn crocus powder solutions resulted in serious chronic effects, including the increased weight loss of colonies, reduced consumption of sugar water and *T. molitor*, and reduced aggressiveness, such as aggregation, grasping ability, and walking speed. RIFAs are well known for their aggressiveness and strong food resource competitiveness [42,43]. The reduced consumption of food and colony expansion as well as their reduced aggregation and grasping ability and walking speed may suggest that another way of controlling RIFAs is to reduce their aggressiveness. Due to the chronic effects, their ability to attack other organisms will be severely handicapped, and their feeding and survival will become difficult, thus affecting the survival of the colony [44]. We did observe an interesting behavior in our experiment. When 10 healthy workers were placed in the same beaker with a *T. molitor*, the workers killed the *T. molitor*. However, when 20 workers, which were fed with sugar water solution containing 100 mg/L crocus bulb powder for 15 days, and were placed in the same beaker with a *T. molitor*, *T. molitor* survived even though the number of workers was doubled in the beaker. This observation suggests that the aggressiveness of the workers fed with sugar water containing 100 mg/L autumn crocus bulb powder were significantly reduced and unable to kill *T. molitor*. 

There is a renewed interest in the use of biopesticides for controlling pests [45,46] due to their biodegradability, less harm to nontarget organisms, and minimum effects on pest resistance [47]. Autumn crocus plant has been cultivated for more than 3000 years, and it is easily produced. The whole plant, leaves, flowers and bulbs contain high concentrations of colchicine [41]. Thus, not only bulbs but also the whole plants could be ground to produce powder, and solutions extracted from the plant tissue could be used for control of RIFAs. The application of the extracts will harm RIFAs and be benign to non-target organisms and the environment due to their biodegradability [48,49]. However, precautions should be taken when use of crocus plants. The plants should be handled by professionals with protection during extraction and application of the extract for RIFA control. 

## 4. Conclusions

This study for the first time documented that solutions made from autumn crocus bulb powder are lethal to RIFAs. The toxicity is mainly attributed to colchicine as RIFAs showed the same response patterns as the commercial colchicine solutions. A sublethal concentration of crocus bulb powder was shown to significantly affect colony growth, food consumption, and aggressiveness of RIFAs, thus reducing their invasiveness. Our study demonstrated that autumn crocus bulb powder could be potentially used as biopesticide for control of RIFAs by either directly killing the RIFAs at a high concentration of crocus bulb powder or by reducing their viability and invasiveness at a lower concentration. Future studies will focus on the mechanisms underlying the toxicity to RIFAs, which may help for the development of valuable botanical pesticides for RIFA control.

## 5. Materials and Methods 

### 5.1. Insects, Chemicals, and Plant

Red imported fire ants (*S. invicta*) were obtained from nests in Guangzhou city, Guangdong Province, China and raised in the laboratory for bioassays. The ants were fed with 10% (*w/w*) sugar water and mealworm. All bioassays were conducted in the laboratory with a temperature of 26 ± 2 °C and a relative humidity of 70%. 

Colchicine standard solution was made using a 97% pure product purchased from Guangzhou Yiyang Trading Co., Ltd. (Guangzhou, China) and stored in a −18 °C freezer in the dark. 

Autumn crocus plants (*C. autumnale*) used in this study were grown at the Insecticidal Botanical Garden at South China Agricultural University, Guangzhou, China (Figure 1A). The bulbs of autumn crocus were collected, washed (Figure 1B), sliced (Figure 1C), dried, and ground into powder (Figure 1D) for the experiments described below.

### 5.2. HPLC Analysis of Colchicine in Autumn Crocus Powder

In reference to Abidin et al. [50] and our preliminary tests, methanol was used for the extraction of colchicine. The colchicine content in the bulb powder of autumn crocus was determined by HPLC. Briefly, 20 g of bulb powder was mixed with 500 mL of methanol in a beaker. After mixing well, the sample was extracted by ultrasonic extraction for 30 min. An amount of 1 mL of the extract was taken and mixed with 50 mL of methanol. The solution was then filtered through a 0.22 μM filter membrane, and the filtrate was used for HPLC analysis of colchicine. A standard curve was established using diluted colchicine standard solutions ranging from 0.1 to 10 mg/L. There were three independent replicates for the extraction and analysis.

HPLC was performed with an Agilent XDB-C_18_ column (250 mm × 4.6 μm × 5 μm) using a mobile phase of methanol–water (45:55), with a flow rate of 1 mL/min, column temperature of 30 °C, detection wavelength of 245 nm, and injection volume of 10 μL.

### 5.3. Toxicity of Colchicine and Autumn Crocus Bulb

The colchicine standard solution was diluted in 10% (*w/w*) sugar water to give concentrations of 50, 30, 10, 5, and 1 mg/L, and autumn crocus bulb powder was dissolved in 10% (*w/w*) sugar water to give doses of 5000, 3000, 1000, 500, and 100 mg/L. Sugar water (10% *w/w*) solution was used as the control. All solutions were freshly prepared.

The experiment was conducted according to the procedures described by Zhang et al. [51]. Ants were starved for 24 h before the experiment. A total of 50 ants were placed in a 1.5 mL test tube and plugged with a cotton ball. The starved ants were subjected to one of the following three treatments: (a) each of the five colchicine concentrations mentioned above, (b) each of the five autumn crocus bulb powder concentrations, and (c) ants were fed with 10% sugar water as the control. The experiment was a completely randomized design with three replications. The number of dead ants was recorded after 3 days of treatment. Additionally, time-course mortality rates of ants fed with sugar water, sugar water containing 1 mg/L colchicine and 100 mg/L autumn crocus bulb powder were recorded every 3 days until day 15. 

### 5.4. Sublethal Effects of Colchicine and Autumn Crocus Bulb Powder on S. invicta Colony Growth

Based on the results of the above toxicity experiment, sugar water solutions containing 1 mg/L colchicine and 100 mg/L autumn crocus bulb powder were selected as sublethal levels for the subsequent experiments. A colony was collected from the field (Guangzhou, Guangdong Province, China) and fed as mentioned above in laboratory conditions (26 ± 2 °C and 70% relative humidity) for one week. The colony was divided into 3 sub-colonies (>25 g) of the same weight, and each sub-colony had alates, broods, workers and at least one functional queen. Sub-colonies were housed in trays (L × W × H, 41.75 × 27.5 × 12 cm) and provided with castone^®^ nests (Diameter × Height, 150 × 25 mm).

The ants in the sub-colonies were starved for 24 h, and then were subjected to three treatments, i.e., fed with (a) 10% (*w/w*) sugar water as the control, (b) 1 mg/L colchicine solution, and (c) 100 mg/L autumn crocus bulb powder solution. The solution was provided in a 15 mL test tube and stoppered with a cotton ball in which enough *T. molitors* was provided for the sub-colonies. The experiment was a completely randomized design with three replications. To quantify the impact of colchicine and autumn crocus bulb powder on *S. invicta* colony growth, each sub-colony was weighed every 3 days up to 15 days after the treatments. The accumulated percentage of weight loss was calculated using the following Equation (1):Weight loss (%) = [(m_2_ − m_1_)/m_2_] × 100%(1)
where m_1_ = the weight of each sub-colony after 0, 3, 6, 9, 12, and 15 days of the treatments and m_2_ = the initial weight of each sub-colony. 

### 5.5. Effects of Colchicine and Autumn Crocus Bulb Powder on Food Consumption of S. invicta 

To determine ant consumption of sugar water and *T. molitors*, sugar water and *T. molitors* were weighed before and after 1, 3, 6, 9, 12 and 15 days of feeding to the fire ant sub-colony. The average consumption (mg sugar water or *T. molitor* per g ant) was calculated for each sub-colony at each sampling day. The experiment was a completely randomized design with three replications. The amount of sugar water evaporated was estimated by weighing three tubes containing each dosage of colchicine and autumn crocus bulb powder before and after they had been kept under the same conditions without the presence of the ants [52].

### 5.6. Effects of Colchicine and Autumn Crocus Bulb Powder on the Aggressiveness of S. invicta 

The aggressiveness of RIFAs was evaluated by three indexes in the experiment, namely aggregating rate, grasping rate, and walking speed of workers. To reduce experimental error, small workers, about 3 mm in length [39,53] were used for recording the walking speed. These three parameters were measured every 3 days up to 15 days of treatment. There were 50 workers per sub-colony. The experiment was a completely randomized design with three replications. Based on the behavior of RIFAs in our research, the aggregation was defined as the gathering of more than five workers. The aggregation rate was calculated using the Equation (2) below:Aggregation rate (%) = (p_1_/p_2_) × 100%(2)
where p_1_ = the number of aggregated workers and p_2_ = the number of total workers measured. 

To test grasping rate, the workers were placed on A4 paper, which was gently rotated 180° after 5 s, and the workers were considered to possess grasping ability if they did not fall from the A4 paper. The following Equation (3) was adopted:Grasping rate (%) = (q_1_/q_2_) × 100%(3)
where and q_1_ = the number of workers possessing grasping ability and q_2_ = the number of total workers measured. 

For evaluation of walking speed, workers were placed on a piece of A4 paper, and the walking distance in 3 s was measured. The walking speed and average walking speed were obtained by the following Equations (4) and (5):Walking speed = Walking distance of workers/3 s(4)
Average walking speed = the sum of all the workers’ walking speed/headcount(5)

### 5.7. Data Analysis

Collected data from the aforementioned experiments were subjected to one-way analysis of variance (ANOVA) using SPSS Statistics, Version 17.0, 2009 (International Business Machines Corporation, Armonk, NY, USA). If significant differences occurred among treatments, means were separated by Tukey’s honestly significant difference (HSD) test at *p* < 0.05 level. Means were presented in graphs with standard error which were drawn using Microsoft Excel.

## Figures and Tables

**Figure 1 toxins-12-00731-f001:**
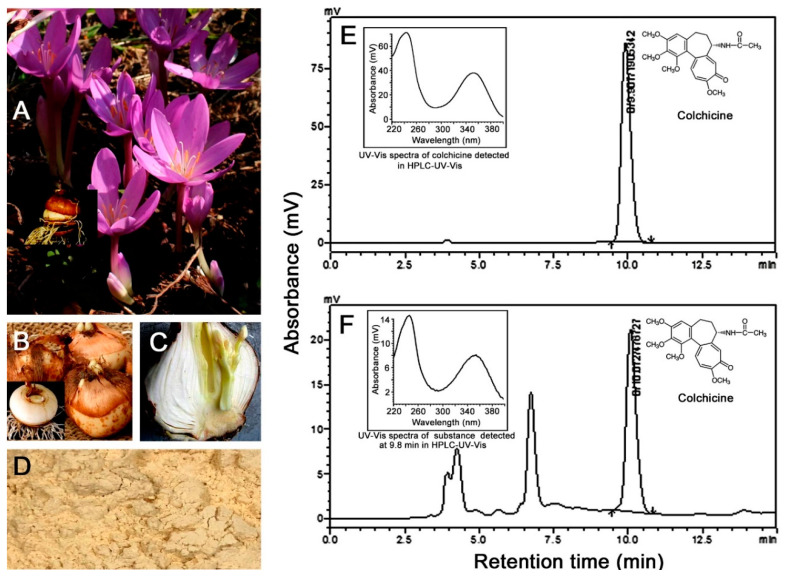
Autumn crocus (*Colchicum autumnale*) plants at the time of flowering (**A**), underground bulbs (**B**), sliced bulb (**C**), ground bulb powder (**D**), HPLC chromatogram of colchicine standard at 10 mg/L (**E**) and autumn crocus bulb powder sample (**F**). The fourth (far right) peak in (**F**) had the same UV-Vis spectra as the peak of colchicine (**E**).

**Figure 2 toxins-12-00731-f002:**
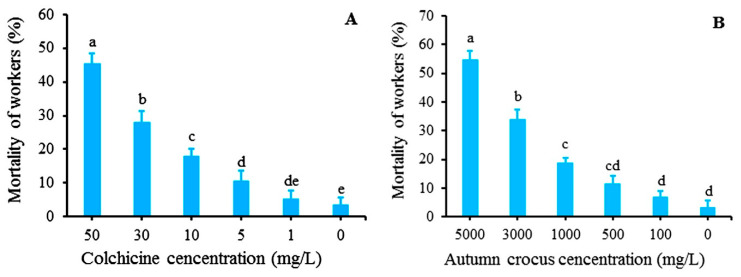
Mortality rates of *S. invicta* ants after three days of feeding with 10% sugar water containing different concentrations of colchicine (**A**) and autumn crocus bulb powder (**B**). The ‘0′ on the x-axis was the control, i.e., 10% of sugar water solution without any addition. Data are presented as mean ± standard error (S.E.). Different letters above bars indicate significant differences in mortality due to concentration effects within a treatment (**A** or **B**) at *p* < 0.05 level based on Tukey’s honestly significant difference (HSD) test (*n* = 3).

**Figure 3 toxins-12-00731-f003:**
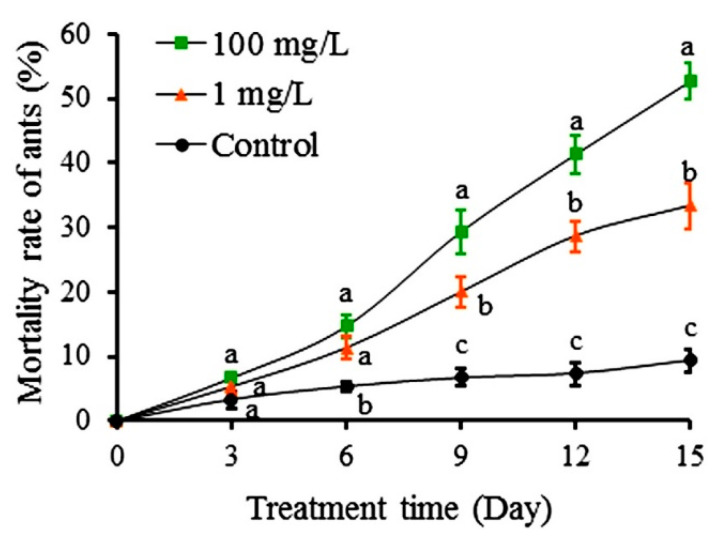
Toxicity of a sublethal concentration of colchicine at 1 mg/L and autumn crocus bulb powder at 100 mg/L in 10% sugar water to *S. invicta* ants over 15 days of feeding. Data are presented as mean ± S.E. Different letters at each observation day indicate significant differences among treatments at *p* < 0.05 level based on Tukey’s HSD test (*n* = 3).

**Figure 4 toxins-12-00731-f004:**
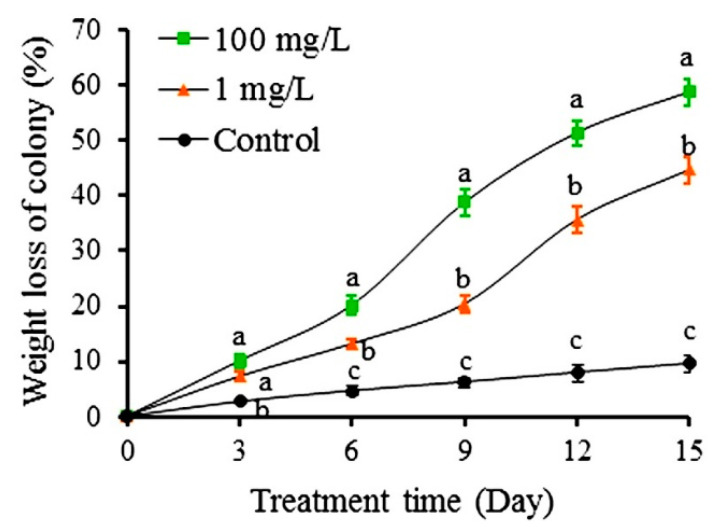
Accumulated colony weight loss (mean ± S.E.) of *S. invicta* colonies over 15 days of feeding with 10% sugar water, the sugar water containing 1 mg/L colchicine, and 100 mg/L autumn crocus bulb, respectively. Different letters at each recording day indicate significant differences among treatments at *p* < 0.05 level based on Tukey’s HSD test (*n* = 3).

**Figure 5 toxins-12-00731-f005:**
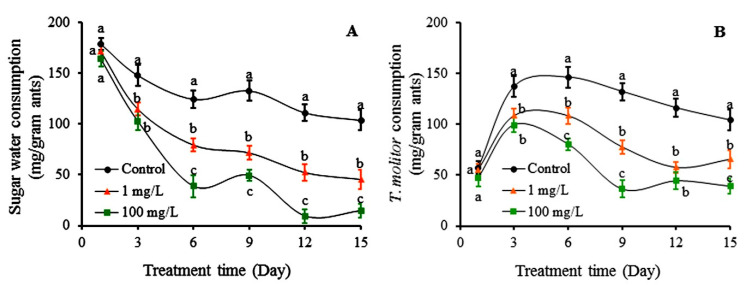
The amount of sugar water (**A**) and *T. molitor* (**B**) consumed by *S. invicta* workers over 15 days of feeding. Data are presented as mean ± S.E. Different letters at each sampling day indicate significant differences among treatments at *p* < 0.05 level based on Tukey’s HSD test (*n* = 3).

**Figure 6 toxins-12-00731-f006:**
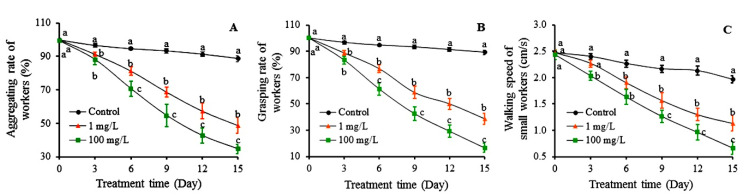
The mean aggregating rate (**A**), grasping rate (**B**), and walking speed (**C**) of small workers of *S. invicta* fed with sugar water and sugar water containing 1 mg/L colchicine or 100 mg/L autumn crocus bulb. Data are presented as mean ± S.E. Different letters at each sampling day indicate significant differences per parameter among treatments at *p* < 0.05 level based on Tukey’s HSD test (*n* = 3).

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
