# Peer review of "Toxicity and Sublethal Effects of Autumn Crocus (Colchicum autumnale) Bulb Powder on Red Imported Fire Ants (Solenopsis invicta)"

_toxins, 2020, doi:10.3390/toxins12110731_

Round 1

Reviewer 1 Report

The presented work is interesting from the point of view of biological experiments on ants, but it is unsuccessful in terms of methodology and results presentation. First of all, it is not clear why the authors use only methanol extraction in experiment for composition determination of dry bulb powder of autumn crocus. How many compounds may be extracted by water solution or organic solvent (for example acetone)? According to figure 1f, 4 main components are extracted, but only one is determined by the authors – colchicine by retention time on HPLC. It is necessary to measure the mass of all components of bulb powder and calculate mass difference measured vs. theoretical for colchicine. Otherwise extracted compound may be close analog but not colchicine itself. Further, the calculation of % colchicine in powder implies its complete extraction by methanol from the sample. It is necessary to prove colchicine complete extraction into methanolic solution (not remain in the pellet), since the content of 0.3% by weight seems to me underestimated according to biological effect of the powder to colchicine alone. I feel that the activity of powder corresponds to the amount of 15% for active component(s). It is also possible that autumn Crocus powder contains several active components with a synergistic effect.

Figure 2 have an error in axis name and shows different units of concentration than results written in the text above. In the caption of all figures, different letters above bars indicate significant differences at P < 0.05 level based on Tukey's HSD test (n = 3). This is complete nonsense and misunderstanding by the authors of the process theory.

The definition for statistical data processing is:

Tukey’s HSD to find out which specific groups’s means (compared with each other) are different. The test compares all possible pairs of means.

The authors do not compare arrays but draw any letters elsewhere.

The number of repeat is minimal; you need to expend up to 5 for the better accuracy of the result.

In figure 3, the mortality rate is deferred, which in my opinion is just mortality without any rate. Further, day after treatment is shown everywhere in figures, it is incorrect since the drugs application continued every day.

A good clear language of presentation and interesting experiments give me pleasure, but serious work is still required to put the material in order.

Author Response

Thank you for reviewing this manuscript and for your constructive comments. We have revised the manuscript based on your suggestions. Here are your comments followed by our response in blue.

1. First of all, it is not clear why the authors use only methanol extraction in experiment for composition determination of dry bulb powder of autumn crocus. How many compounds may be extracted by water solution or organic solvent (for example acetone)?

A purpose of this study was to extract as much colchicine as possible from dry bulb powder of autumn crocus and minimize the extraction of other compounds. Based on a study conducted by Abidin et al (2014) as well as our preliminary studies, methanol was found to be effective for colchicine extraction compared to ethanol, chloroform, and acetone. Thus, methanol was used for extraction in this study.

Abidin, L.; Khurana, D.; Mujeeb, M. 2014. Effect of process parameters on the extraction of colchicine from Colchicum autumnale L. seeds. BMR Biotechnol. 2014, 1, 1-5.

2. According to figure 1f, 4 main components are extracted, but only one is determined by the authors – colchicine by retention time on HPLC. It is necessary to measure the mass of all components of bulb powder and calculate mass difference measured vs. theoretical for colchicine. Otherwise extracted compound may be close analog but not colchicine itself. 

Crocus plants are rich in colchicine, and methanol is effective for colchicine extraction. There were four peaks appeared from the crocus powder extract. Our attention was focused on colchicine, i.e., the last peak, which had the same retention time as the peak of the colchicine reference (Fig. 1E). To confirm if the last peak was colchicine, we analyzed the UV-Vis spectra of colchicine in the standard sample by HPLC-UV-Vis, and then measured the UV-Vis spectra of the fourth peak of crocus powder extract. The UV-Vis spectra of the two samples were consistent, which suggested that the two were the same compound. In this revision, we have added the UV-Vis spectra in Figure 1E and F, respectively. The third peak showed the similar UV-Vis spectra as colchicine, we assume it could be colchicoside. The first and second peaks appeared together, which require further investigation. 

3. Further, the calculation of % colchicine in powder implies its complete extraction by methanol from the sample. It is necessary to prove colchicine complete extraction into methanolic solution (not remain in the pellet), since the content of 0.3% by weight seems to me underestimated according to biological effect of the powder to colchicine alone. I feel that the activity of powder corresponds to the amount of 15% for active component(s). It is also possible that autumn Crocus powder contains several active components with a synergistic effect.

To verify the extraction efficiency by methanol, we use acetone and hexane to extract colchicine from pellets. Results showed that colchicine contents in the pellet were less than 2% and 5% of the initial extraction with methanol, which indicated that most of the colchicine in dry bulb powder of autumn crocus had been extracted by methanol. We agree with you that under the same colchicine concentration, the activity of extract derived from bulb powder was higher than that of colchicine standard, which may be due to the synergistic effect of other substances, such as colchicoside.

4. Figure 2 have an error in axis name and shows different units of concentration than results written in the text above.

Thank you for pointing it out. We have revise it.

5. In the caption of all figures, different letters above bars indicate significant differences at P < 0.05 level based on Tukey's HSD test (n = 3). This is complete nonsense and misunderstanding by the authors of the process theory.

The definition for statistical data processing is:

Tukey’s HSD to find out which specific groups’s means (compared with each other) are different. The test compares all possible pairs of means.

The authors do not compare arrays but draw any letters elsewhere.

We have revised the captions by stating more specifically.

6.In figure 3, the mortality rate is deferred, which in my opinion is just mortality without any rate. Further, day after treatment is shown everywhere in figures, it is incorrect since the drugs application continued every day.

We have changed the abscissa title to “Treatment time (Day)”.

7. The number of repeat is minimal; you need to expend up to 5 for the better accuracy of the result.

The standard errors of all data were small, so the number of repeat was minimal. We should be aware of such concern in our future experiments.

Reviewer 2 Report

Manuscript ID: toxins-953481
Type of manuscript: Article
Title: Toxicity and Sublethal Effects of Autumn Crocus (Colchicum autumnale)
Bulb Powder on Red Imported Fire Ants (Solenopsis invicta)

Comments to Editor and authors

In the manuscript entitled “Toxicity and Sublethal Effects of Autumn Crocus (Colchicum autumnale) Bulb Powder on Red Imported Fire Ants (Solenopsis invicta)” the authors describe the use of Autumn Crocus bulb powder  as insecticide against Solenopsis invicta  

Major revision:The authors on page 2 (par 2.1) describe the procedure for estimating the concentration of colchicine in bulb powder via retention time in a chromatographic system. It would be appropriate to characterize the chromatographic peak with mass spectrometry to unequivocally demonstrate the presence of colchicine. Moreover in figure 1 are not given indications on the first two peaks, not even about contamination molecules.

 In Figure 2 there are several problems. Better to standardize the concentration scales in the X axes, i.e. put both in panel A (concentration of colchicine) and in panel B (concentration of autumn crocus) the concentration units as mg/L To demonstrate the colchicine activity both as standard and autumn crocus powder it would be better to do a control experiment in histology to verify that there is an effective block of the mitotic spindle on the sections of killed fire ants. A further experiment could be performed using powder from crocus waste products: petals, leaves, etc. to enhance the waste in a closed cycle.  

Minor revisions:

  • Page 2 lane 70: please correct 3.2 with 2.2        
  • References, page 10, Lane 379: 2018 please write 2018·        
  • References, page 11, Lane 401: 2009 please write 2009·        
  • References, page 11, Lane 419: (Acorus Calamus)  please write (Acorus calamus)·        
  • References, page 11, Lane 433: Typha angustifolia  please write Typha angustifolia

For these reasons the manuscript in the present form cannot be accepted.

Author Response

I would like to thank you for reviewing this manuscript and for your constructive comments on the improvement of the manuscript. Based on your suggestions, we have revised the manuscript. Here is a list of your comments followed by our responses.

1. The authors on page 2 (par 2.1) describe the procedure for estimating the concentration of colchicine in bulb powder via retention time in a chromatographic system. It would be appropriate to characterize the chromatographic peak with mass spectrometry to unequivocally demonstrate the presence of colchicine. Moreover, in figure 1 are not given indications on the first two peaks, not even about contamination molecules.

Thank you for the suggestion. Crocus plants are rich in colchicine, and methanol is effective for colchicine extraction (Abidin et al., 2014). There were four peaks appeared from the crocus powder extract. Our attention was focused on colchicine, i.e., the last peak, which had the same retention time as the peak of the colchicine reference (Fig. 1E). To confirm if the last peak was colchicine, we determined the UV/Vis spectra of colchicine in the standard sample by HPLC-UV-vis and the UV/Vis spectra of the fourth peak of crocus powder extract. The UV-vis spectra of the two samples were consistent, suggesting that the two were the same compound. In this revision, we have added the UV-Vis spectra in Figure 1E and F, respectively. The third peak showed the similar UV-vis spectra as colchicine, we assume it could be colchicoside, a derivative of colchicine. The first and second peaks appeared together, which require further investigation. Our future research will characterize those peaks as your suggested.

2. In Figure 2 there are several problems. Better to standardize the concentration scales in the X axes, i.e. put both in panel A (concentration of colchicine) and in panel B (concentration of autumn crocus) the concentration units as mg/L. To demonstrate the colchicine activity both as standard and autumn crocus powder it would be better to do a control experiment in histology to verify that there is an effective block of the mitotic spindle on the sections of killed fire ants. A further experiment could be performed using powder from crocus waste products: petals, leaves, etc. to enhance the waste in a closed cycle. 

The concentration scales in X axes have been changed to mg/L. Thank you for your suggestions on the direction of future research. As this is the first report on lethal effects of colchicine on fire ants, future studies are warranted regarding mechanisms underlying the lethal effects. Colchicine blocks mitotic spindle of fire ants could be one of mechanisms. As you mentioned, extracts from other organs of crocus, such as petal, leaves, and seeds will also be evaluated. Again, thank you for your suggestions.

3. Page 2 lane 70: please correct 3.2 with 2.2        

    References, page 10, Lane 379: 2018 please write 2018        

    References, page 11, Lane 401: 2009 please write 2009       

    References, page 11, Lane 419: (Acorus Calamus)  please write (Acorus calamus)        

References, page 11, Lane 433: Typha angustifolia  please write Typha angustifolia

 All suggested changes have been made.

Reviewer 3 Report

In my opinion the methodology is not well explained. 

As I am not specilist in statistics I did not check the analyses, shoulkd be checked by an statistician. 

Author Response

I would like to thank you for reviewing our manuscript and for your constructive comments. Based on your suggestions, we have revised the manuscript. Here is a list of your comments followed by our responses.

1. Title should be "Toxicity and sub-lethal effects of commercial colchicine and autumn crocus (Colchicum autumnale) bulb powder on red imported fire ants (solenopsis invicta)" because authors used commercially available colchicine.

Thanks for the suggestion. This study for the first time demonstrated that extract derived from crocus bulb powder was toxic to fire ants, and the toxic effect was mainly attributed to colchicine. We feel that commercial colchicine used in this study as a positive control. So, we did not modify the title.

2. About the use of sub-lethal and sublethal in the manuscript.

We have change to sublethal throughout.

3. Please explain what you mean by "short-term feeding" how many hours or days?

It was 15 day, which was explained in the subsequent sentence.

4. What do you mean "colony growth" Do you mean "numbers of egg laying, larval development, pupae and new adults?

"Colony growth" means the change of the colony weight. We have made it clear in this revision.

5. Please explain what you mean? "balance of natural ecosystems"

We have revised this sentence. Red fire ants can damage plants and bite various creatures, and they have no natural enemies. Due to the invasiveness, they can break existing natural ecosystem.

6. Line 75, How many different concentrations you used? please mention here.

This sentence has been modified.

7. You should explain the testing and feeding method. did you use open containers for feeding?

The testing and feeding method was described in 5.3, "The experiment was conducted according to the procedures described by Zhang et al. [51]. Ants were starved for 24 hours before the experiment. A total of 50 ants were placed in a 1.5 mL test tube and plugged with a cotton ball."

8. How you measured the consumption of the ants feeding on the mealworm? What do you mean "per gram workers"? Do you mean "per gram per worker"? Please explain.

In 5.5 (Effects of colchicine and autumn crocus bulb powder on food consumption of S. invicta),  the sugar water or T. molitors were weighed before and after each time point, and the ants at each time point were also weighed (see 5.4), then divide the consumption of sugar water or T. molitors by the weight of the ants to derive food consumption rate (mg sugar water or T. molitors/g ants).

9. What do you mean "small workers" did colony had different sizes of workers?

As described in 5.4 and 5.6, there are different types of ants in the colony, like alates, broods, workers, and queen. Workers have different sizes, and the walking speeds between big and small workers are different. Hence, only small workers, about 3 mm in length were used for recording the walking speed to reduce experimental error.

10. Other suggestion: Line 7 “ground” should be “grounded”; Line 9 “ant” should be “ants”, Line 10 “in” should be “within”, Line 33 “was” should be “is”, Line 34 “rapidly” should be “found”, Line 40 “drug” should be “insecticides”, Line 41 “low-resistant” should be “may be inherited tolerance”, Line 47 insert "synthetic", Line 78 insert "mortality", Line 85 “ground” should be “grounded”, Line 193 “enormous” should be “major”, Line 189 “Lard et al.” should be “Lard C.F.”, Line 265 “purity” should be “pure”, Line 269 “ground” should be “grounded”.

Thank you again for pointing out the problems. All suggested changes have been made except for “ground”. The past participle of grind is ground.

Reviewer 4 Report

This is a review of the paper entitled “Toxicity and sublethal effects of autumn crocus (Colchicum autumnale) bulb powder on red imported fire ant (Solenopsis invicta). Red imported fire ant is a serious pest to both agriculture and urban environments. Improved management techniques for this pest would be welcome. The results are very clean and there is no need for additional speculation on the biodegradability or environmental friendliness of the product. There is a risk that people will equate “natural” with “safe.” While colchicine is used as a drug, the concentrations needed for use as an insecticide may be harmful and dangerous if used inappropriately. How much of the powder would I have to breathe to get a lethal dose?

Line 21) I am not sure how the authors are justifying “environmentally and ecologically benign.” Natural does not equate with benign. Cyanide, ricin, nicotine, and many others are all natural. There are also hybrids where pyrethrum is a natural insecticide while pyrethroid (line 38) is a chemical derivative of pyrethrum.

Line 88) The assays are both lethal and sublethal. The last half of the paragraph is on mortality.

Line 170) … where the control treatment …

Line 171 … than the rest of the treatments.

Line 200) For example, the essential …

Line 206) Is this practice safe for the people? No gloves, no masks, mix up a bit, taste, and spray?

Line 211) I strongly dislike mixing terms to please a wide audience, and “bulb or corm” is terrible scientific writing. That said, there are people who feel strongly that this is a bulb or this is a corm. Thankfully your figure 1C clearly shows that there are distinct layers. The structure looks like an onion. The next part is to find the scientific definition for bulb/corm. I used my 1982 textbook from my introductory botany class “Botany: an introduction to plant biology 6th ed by Weier et al.). “Corm (Gr. Kormos, a trunk) a short, solid, vertical, enlarged underground stem in which food is stored.” The key is solid, and your picture clearly shows that the structure is not solid. The definition of bulb is “… with many fleshy scale-like leaves filled with stored food.” The description for bulb fits your picture, and I conclude that the persons who are calling this a corm are misinformed or delusional. In my world bulbs and corms are different anatomical structures and they are not the same. The bulb is a modified bud, the corm is a modified stem.

     If you can find a taxonomist/systematist skilled in identification of plants (I study insects) that tells you that the definition has changed and you have a corm, then I can accept that. In either case only use one word. Thank you for Figure 1C.

Line 225) Did you see them attack each other? No data were provided for such.

Line 226) the survival of the colony.

Line 230) I think the subject is “the aggressiveness” so line 232 should be “was significantly reduced”

Line 232) … reduced, and the ants were unable …

Line 233) The use of biopesticides is not new in terms of the history of modern pest management. So if biopesticides are “new” then so are most of the synthetic pesticides. Nicotine, neem, B.t., spinosyns are a few of the natural biopesticides that have been around for some time.

Biopesticides are environmentally friendly when used at rates found in nature. We extract and concentrate them, and the result is a pesticide that has many of the problems of conventional pesticides. It might be quite hazardous to humans and wildlife to start growing and using this material. I suspect that there should be some disclaimer to discourage people from trying this at home using pots and pans in the kitchen.

Line 248) .. at lower concentrations.

Author Response

Thank you so much for your constructive comments on the improvement of this manuscript. Based on your suggestions, we have revised the manuscript. Here is a list of your comments followed by our responses.

1. There is a risk that people will equate “natural” with “safe”. While colchicine is used as a drug, the concentrations needed for use as an insecticide may be harmful and dangerous if used inappropriately. How much of the powder would I have to breathe to get a lethal dose? Line 21) I am not sure how the authors are justifying “environmentally and ecologically benign”. Natural does not equate with benign. Cyanide, ricin, nicotine, and many others are all natural. There are also hybrids where pyrethrum is a natural insecticide while pyrethroid (line 38) is a chemical derivative of pyrethrum.

We completely agree with you that not all natural products are safe. Although colchicine has been used clinically to treat cancer and gout as a drug, it is extremely toxic at high concentrations.  Based on your comments, we have modified relevant sentences in this revision.

2. Line 206) Is this practice safe for the people? No gloves, no masks, mix up a bit, taste, and spray?

Cautions should be taken when handling crocus bulbs. It is recommended to wear gloves and use face masks, and absolutely no ingestion of bulbs.

3. Line 211) I strongly dislike mixing terms to please a wide audience, and “bulb or corm” is terrible scientific writing. That said, there are people who feel strongly that this is a bulb or this is a corm. Thankfully your figure 1C clearly shows that there are distinct layers. The structure looks like an onion. The next part is to find the scientific definition for bulb/corm. I used my 1982 textbook from my introductory botany class “Botany: an introduction to plant biology 6th ed by Weier et al.)”. “Corm (Gr. Kormos, a trunk) a short, solid, vertical, enlarged underground stem in which food is stored”. The key is solid, and your picture clearly shows that the structure is not solid. The definition of bulb is “… with many fleshy scale-like leaves filled with stored food”. The description for bulb fits your picture, and I conclude that the persons who are calling this a corm are misinformed or delusional. In my world bulbs and corms are different anatomical structures and they are not the same. The bulb is a modified bud, the corm is a modified stem.

If you can find a taxonomist/systematist skilled in identification of plants (I study insects) that tells you that the definition has changed and you have a corm, then I can accept that. In either case only use one word. Thank you for Figure 1C.

Thank you very much for your comments. We agree with you that ‘bulb’ should be used. We have corrections.

4. Line 225) Did you see them attack each other? No data were provided for such.

The sentence was deleted. 

5. Line 230) I think the subject is “the aggressiveness” so line 232 should be “was significantly reduced”.

The suggested change has been made.

6. Line 233) The use of biopesticides is not new in terms of the history of modern pest management. So if biopesticides are “new” then so are most of the synthetic pesticides. Nicotine, neem, B.t., spinosyns are a few of the natural biopesticides that have been around for some time.

Biopesticides are environmentally friendly when used at rates found in nature. We extract and concentrate them, and the result is a pesticide that has many of the problems of conventional pesticides. It might be quite hazardous to humans and wildlife to start growing and using this material. I suspect that there should be some disclaimer to discourage people from trying this at home using pots and pans in the kitchen.  

Thank you for the suggestion. Appropriate revisions have been made.

Round 2

Reviewer 1 Report

I am not satisfied by author’s response to my first-round revision. I believe that without molecular weights determination of all components eluted in figure 1F and without proper statistical processing of data on figure 2, 3, 4, 5 and 6 (like rules and the presentation practice for science), the article cannot be published.

Reviewer 2 Report

After the corrections/additions made and after having provided the necessary explanations, the work can be accepted in the present form